# Advances in Antimicrobial Peptide Discovery via Machine Learning and Delivery via Nanotechnology

**DOI:** 10.3390/microorganisms11051129

**Published:** 2023-04-26

**Authors:** Alexa Sowers, Guangshun Wang, Malcolm Xing, Bingyun Li

**Affiliations:** 1Department of Orthopaedics, School of Medicine, West Virginia University, Morgantown, WV 26506, USA; 2School of Pharmacy, West Virginia University, Morgantown, WV 26506, USA; 3Department of Pathology and Microbiology, College of Medicine, University of Nebraska Medical Center, 985900 Nebraska Medical Center, Omaha, NE 68198, USA; 4Department of Mechanical Engineering, University of Manitoba, Winnipeg, MB R3T 2N2, Canada

**Keywords:** antibiotic resistance, antimicrobial peptide, drug delivery, LL-37, machine learning, nanotechnology, peptide engineering, peptide database, peptide design

## Abstract

Antimicrobial peptides (AMPs) have been investigated for their potential use as an alternative to antibiotics due to the increased demand for new antimicrobial agents. AMPs, widely found in nature and obtained from microorganisms, have a broad range of antimicrobial protection, allowing them to be applied in the treatment of infections caused by various pathogenic microorganisms. Since these peptides are primarily cationic, they prefer anionic bacterial membranes due to electrostatic interactions. However, the applications of AMPs are currently limited owing to their hemolytic activity, poor bioavailability, degradation from proteolytic enzymes, and high-cost production. To overcome these limitations, nanotechnology has been used to improve AMP bioavailability, permeation across barriers, and/or protection against degradation. In addition, machine learning has been investigated due to its time-saving and cost-effective algorithms to predict AMPs. There are numerous databases available to train machine learning models. In this review, we focus on nanotechnology approaches for AMP delivery and advances in AMP design via machine learning. The AMP sources, classification, structures, antimicrobial mechanisms, their role in diseases, peptide engineering technologies, currently available databases, and machine learning techniques used to predict AMPs with minimal toxicity are discussed in detail.

## 1. Introduction

The discovery of antibiotics is one of the greatest achievements in modern medicine. However, over time, antibiotics have become less effective due to the emergence of drug-resistant bacteria from the overuse and abuse of antibiotics [1]. In clinical isolates, antibiotic resistance mutations have been found in promoter regions of efflux pumps, the target of antibiotics, and binding regions of antibiotics. Antibiotic resistance, where a mutation or post-translational modification alters the target or inactivates the antibiotic [2,3], has resulted in treatment failure [3]. The Centers for Disease Control and Prevention (CDC) reports more than 2.8 million infections and over 35,000 deaths due to antibiotic resistance in the United States [4]. Therefore, discovering new antimicrobial agents to overcome this challenge is essential [5].

Antimicrobial peptides (AMPs) constitute an important source for developing a new generation of antibiotics. AMPs have a broad range of activity against microorganisms, including bacteria [6], fungi [7], parasites [8], and viruses [9,10]. These peptides play a critical role in innate immunity by responding to a variety of pathogen-associated molecular patterns through increased leukocyte recruitment to the infection site and signaling damaged tissues [11,12]. AMPs also are involved in the adaptive immune response as they have demonstrated the ability to recruit immature phagocytic and dendritic cells [13]. AMPs may exhibit other properties, including the ability to target drug-resistant microbial biofilms, kill cancer cells, and promote wound healing [14]. Biofilm inhibition is an advantageous property of AMPs, as biofilms have led to increased tolerance to various antibiotics and have accounted for about two-thirds of all human infections [13]. In the case of chronic wounds, the formation of biofilms could delay wound healing. Hence, the effective disruption of preformed biofilms by AMPs constitutes an important step toward wound healing [15,16,17,18].

AMPs have been discovered in six life kingdoms: bacteria, archaea, protists, fungi, plants, and animals [19]. Most natural AMPs (74%) originated from animals, especially amphibians and insects [20,21]. They share some common characteristics, such as cationic charge (+1 to +7), short sequences (<50 amino acids), and they are usually amphiphilic [13,22]. The net positive charge results from the abundance of arginine, lysine, and histidine residues due to the positive charges associated with these residues [23]. The presence of ~50% hydrophobic and hydrophilic residues contributes to these peptides’ amphiphilicity, enabling them to interact with bacterial membranes and to enter cells [13]. These peptides can fold into different structures, including alpha helices, beta-sheets, combined alpha helix and beta-sheets, and non-alpha-beta structures [24]. Amphipathic alpha-helical structures, observed for magainin and human cathelicidin LL-37 for instance, provide a basis for understanding the mechanism of action of AMPs [25,26,27].

AMPs present selectivity toward bacteria over eukaryotic cells. AMPs’ selectivity toward bacterial membranes over eukaryotic membranes is due to differences in the composition of the membranes. Eukaryotic membranes have primarily zwitterionic lipids, including phosphatidylcholine and sphingomyelin, which are usually neutral. In contrast, bacterial membranes are composed of anionic lipids, such as phosphatidylglycerol and cardiolipin [28]. The favored interaction between cationic AMPs and anionic bacterial membranes is responsible for peptide selectivity.

It should be pointed out that such a preferred interaction with bacteria does not mean AMPs will never interact with host cells. AMPs, such as human cathelicidin and defensins, do associate with host cell receptors (e.g., GPCRs, MrgX2, FPR2, and P2X7) at a peptide concentration much below the minimal inhibitory concentration (MIC) [29]. Such interactions are not cell damaging; rather, they induce signal transduction for immune regulation. Such a role of AMPs links them with a variety of physiological processes, including human diseases.

## 2. AMPs and their Relationships with Human Diseases

### 2.1. Role in Respiratory Diseases

AMPs provide defense mechanisms for multiple diseases, including respiratory diseases. Rai et al. reviewed the preparation of various AMP-based materials and their antimicrobial properties against infections in the brain, eye, mouth, skin, lung, etc. [30]. Due to their role in adaptive and innate immunity, AMPs can protect against pathogens in various organs [31]. The respiratory tract is lined with epithelial cells to protect against potential infections [13]. Upon infection, epithelial cells, neutrophils, and macrophages populate and can secrete AMPs, which act intracellularly to kill invading pathogens [32]. Cathelicidin LL-37, a natural human AMP, and cathelicidin-related AMP (CRAMP) demonstrated antiviral activity against influenza A virus [33,34]. LL-37 showed effectiveness against influenza A virus both in vitro and in vivo, with a 90% reduction in virus titer in vitro and a significant 70–80% reduction in lung virus three days post-infection in mice [34]. AMPs have demonstrated their role in protecting against respiratory infections, but they are also involved in non-infectious lung diseases [32].

In various respiratory diseases, such as chronic obstructive pulmonary disease (COPD) and cystic fibrosis, studies have presented different AMP expression (overexpression or down-regulation) compared to non-diseased states [13]. For instance, some AMPs, such as LL-37, are overexpressed in respiratory diseases, which may influence disease progression. Increased levels of LL-37 correlated with bronchial inflammation, which affected disease severity [35]. The overexpression of LL-37 in the airways of COPD patients stimulated the overproduction of mucus, where this mucus contributed to disease progression [36]. LL-37 is a neutrophil-derived AMP, which can further disease progression in the lungs due to its inflammatory activity and cytotoxicity at relatively high concentrations [32]. Smokers with COPD, a disease with limited airflow due to toxic particles or gases, had reduced beta defensin-2 (hBD-2) compared to smokers without COPD and ex-smokers with COPD [37,38]. Decreased concentrations of hBD-2 were also associated with disease severity in cystic fibrosis patients [35]. Diseases involving decreased hBD-2 production could correspond to an increased susceptibility to infections by different pathogens [13].

### 2.2. Role in Autoimmune Diseases

Studies have shown that the expression of AMPs plays a role in autoimmune disease pathogenesis. Psoriasis, rheumatoid arthritis, and Crohn’s disease are autoimmune diseases that have shown altered AMP expression [39,40,41]. Psoriasis is an autoimmune skin condition caused by innate and adaptive immunity deviations induced by infection, injury, and stimulation [42]. Patients diagnosed with psoriasis have increased expression levels of AMPs, such as LL-37 and human β-defensins, compared to healthy individuals [43]. Studies demonstrated that LL-37 could act as an autoantigen by circulating T cells and can complex with self-RNA, which activates Toll-like receptors (TLR), leading to the exacerbation of psoriasis [44,45]. hBDs also play a role in the development of psoriasis as Rohrl et al. demonstrated the ability of hBD-2 to act as a ligand for chemokine receptor (CCR6), where this CCR6 signal induced Th17 in patients with psoriasis [46,47].

Rheumatoid arthritis is a disease that leads to chronic inflammation and joint destruction by producing disease-specific-anticitrullinated protein antibodies and complexing with citrullinated fibrinogen [48]. Francisco et al. discovered elevated liver-expressed AMP 2 (LEAP2) levels in rheumatoid arthritis patients compared to healthy individuals. This evidence indicates that LEAP2 is associated with inflammatory mediators in rheumatoid arthritis [49]. hBD-3 also played a role in the pathogenesis of rheumatoid arthritis, indicated by the destruction of the cartilage due to the activation of matrix metalloproteinases, resulting in the degradation of the extracellular matrix cartilage [50,51].

Crohn’s disease is a chronic condition described by relapsing and remitting patchy inflammation along any section of the gastrointestinal tract [52]. Gutiérrez et al. discovered the upregulated expression of LL-37 in peripheral blood neutrophils in Crohn’s disease patients [53]. Another study demonstrated increased LL-37 expression in the inflamed mucosa of patients with Crohn’s disease. In Crohn’s disease, LL-37 may enhance antibacterial properties, specifically in the inflamed mucosa, which may protect against infection [54]. Wehkamp et al. determined that Crohn’s disease patients have reduced antibacterial activity in the intestinal mucosa and have shown a decreased expression in human α-defensin 5 (HD5) in the ileum. This decreased α-defensin expression may be a factor in the pathogenesis of Crohn’s disease through inflammation in the ileum, leading to compromised innate immunity [55].

Altered expression levels of AMP in the gastrointestinal tract have been found in diabetes. Liang et al. determined that cathelicidin-related antimicrobial peptide (CRAMP) expression was decreased in the colon of non-obese mice, which led to the production of type I interferons. This production led to the promotion of the pancreatic autoimmune response, accelerating the progression of diabetes. However, when CRAMP expression was increased, it restored homeostasis, which prevented the diabetogenic response [56]. Although type 2 diabetes is not an autoimmune disease, this disease has been associated with a reduced expression of β-defensin [57]. Lin et al. determined that hBD-1, human beta defensin 1, was reduced in the gastric corpus in the diabetic metabolic state. Given this information, the induction of AMP expression may help resolve issues involved in intestinal barrier deficiency [58].

### 2.3. Role in Cancer

Vitamin D receptor (VDR) plays a role in regulating the transcriptional profile of many genes involved in physiological functions, such as regulating immune activity [13]. VDR has been shown to be one of the resistance mechanisms to pathogens. Due to the dysregulation of the VDR, AMP expression levels alter, and cancer biology is influenced through altered DNA methylation patterns [59]. Various AMPs, such as α-defensins, β-defensins, and LL-37, are involved in developing a variety of tumors and cancers, where the VDR regulates these peptides [13]. Defensins are AMPs produced by eukaryotes, where some defensin-like peptides have antiproliferative activity and apoptosis in cancer cells, demonstrated by the increased phosphorylation of MAPK p38 [60,61]. Due to this ability, defensins can be used in combination therapy to overcome chemotherapeutic resistance. For instance, Johnstone et al. determined that defensins could increase the in vitro anticancer activity of doxorubicin, an anticancer drug, against multidrug-resistant tumor cells [62]. Human neutrophil peptide-1 (HNP-1) is another alpha defensin that has shown anticancer activity in addition to antimicrobial activity [63]. Gaspar et al. established that HNP-1 induced apoptosis with defects in the cellular membrane at a low concentration of peptide [64].

Likely due to the exposure of anionic phosphatidylserine (PS), cancer cells have an increased sensitivity to cationic AMPs compared to normal cells. This increased sensitivity results from an undeveloped cytoskeleton seen in cancer cells [1]. Cancer cells have a high metabolism, which allows this change in the cytoskeleton. As a result, AMPs can interact with the cancer cell membrane [1,13]. Interestingly, LL-37 has demonstrated the ability to produce tumorigenic or anticancer effects, depending on the type of cancer involved [65]. For instance, the expression of LL-37 is down-regulated in gastric and pancreatic cancer, indicating that LL-37 can have anticancer effects. Zhang et al. showed a high concentration of LL-37 at 20 mg/kg with a 42% reduction in pancreatic tumor growth compared to the control. Results have suggested that the LL-37-induced inhibition of autophagy results in the accumulation of reactive oxygen species (ROS) and inhibits pancreatic cancer cell growth [66]. Wu et al. determined that LL-37 inhibited the proliferation of gastric cancer cells by activating bone morphogenetic protein signaling through a proteasome-dependent mechanism [67]. Of great interest is that the major antimicrobial peptide (FK-16) of LL-37, originally discovered by NMR-trim, is demonstrated to have anticancer properties as well [68,69].

LL-37 has also been shown to present tumorigenic effects in some cases. Haussen et al. showed LL-37 expression in human lung cancer cells; the overexpression of LL-37 in cancer cells resulted in an increased proliferation. This study also showed that murine models had similar results, where cells overexpressing LL-37 developed larger tumors compared to mice without the overexpression of LL-37 [70]. The overexpression of cathelicidins has also played a role in the pathogenesis of breast cancer. Weber et al. determined that the treatment of LL-37 stimulated breast cancer cell migration and their colonies had a dispersed morphology, which indicated increased metastatic potential. The overexpression of human cathelicidin protein (hCAP18) in low malignant breast cancer cell lines induced metastases in severe combined immunodeficiency mice. Therefore, the overexpression of hCAP18 and LL-37 in certain cancers may serve as a useful marker for cancer diagnosis [71].

### 2.4. Role in Cardiovascular Diseases

Cardiovascular disease contributes to worldwide mortality, including conditions such as atherosclerosis and heart failure [72]. Atherosclerosis is the accumulation of plaque in blood vessels and is one of the leading causes of cardiovascular diseases [73]. Edfeldt et al. discovered the expression of LL-37 in atherosclerotic lesions, which modulates inflammation expression. LL-37 activated adhesion molecules and chemokines, resulting in leukocyte recruitment and atherogenesis [74,75]. Salamah et al. determined the presence of LL-37 in platelets, which secreted upon activation and promoted thrombus formation. The formation of a thrombus led to further complications, such as blood clots, resulting in heart failure [76].

Heart failure is an end-stage condition that results from complications associated with cardiovascular disease, which is identified as the leading cause [77]. Zhou et al. demonstrated the decreased expression of LL-37 in heart failure patients and decreased CRAMP in heart and serum samples of heart failure mice models. In their study, CRAMP supplementation suppressed cardiac hypertrophy, indicated by reduced expression levels of atrial natriuretic peptide and B-type natriuretic peptide. The presence of LL-37 in serum suggested that this AMP might be used as a biomarker for acute heart failure [78]. For instance, Bei et al. demonstrated that a lower LL-37/neutrophil ratio could predict a worse prognosis for myocardial infarction patients [79].

### 2.5. Role in Neurodegenerative Diseases

Neuroinflammation results from brain injuries or diseases released from activated microglia, astrocytes, and cytokines. Neurodegeneration is associated with responses due to inflammation resulting from over-activated glial cells [80]. Lee et al. showed a relative upregulation of LL-37 in the brain compared to other organs, indicating that the LL-37 expression level might be associated with chronically diseased areas in the brain involved in Alzheimer’s and Parkinson’s [80]. The amyloid β-protein (Aβ), another human AMP, influences Alzheimer’s disease pathogenesis [81]. Soscia et al. determined that Alzheimer’s disease homogenates increased higher antimicrobial activity against microbes compared to non-diseased homogenates. Although Aβ demonstrated its ability to protect Alzheimer’s patients from infection, this peptide also contributed to the pathogenesis of the disease [81]. Wang et al. determined that dysregulated AMP expression might be involved in the development of Alzheimer’s disease by inducing Aβ deposition. These findings indicated the potential of using AMPs as biomarkers and therapeutic agents for Alzheimer’s disease [82].

## 3. AMPs’ Antimicrobial Mechanisms

AMPs kill microorganisms primarily via selectively disrupting the membrane through membrane permeation mechanisms. These mechanisms include the barrel-stave model, carpet model, and toroidal-pore model (Figure 1) [28]. In the barrel-stave model, AMPs are embedded perpendicularly into the microbial membrane, where they self-assemble to form a transmembrane channel [83,84]. Due to their amphipathic nature, the hydrophobic faces orient themselves toward the interior of the microbial membrane, while the hydrophilic faces orient themselves to form the hydrophilic pore (Figure 1b) [85]. In the carpet model, AMPs are oriented parallel to the microbial membrane, where they are absorbed to induce membrane disruption. The membrane disrupts through micellization, degrading the membrane and resulting in microbial cell death, as shown in Figure 1c [86]. The toroidal-pore model is similar to the barrel-stave model, where a pore is formed by inserting AMPs perpendicularly into the membrane. However, this model differs from the barrel-stave model where the bacterial membrane incorporates into the pore. This incorporation induces lipid monolayers to bend, forming a continuous interface with the AMPs (Figure 1d) [28,84]. Generally, AMPs target microbial membranes through hydrophobic and electrostatic interactions, resulting in cell death due to losing cellular components and electrochemical gradients [87,88]. However, evidence suggests that membrane permeation is not the only mechanism for AMPs to eliminate bacteria. Some AMPs, such as lantibiotics, target the bacterial cell wall [89]. AMPs may also translocate across microbial membranes and bind to intracellular targets that may inhibit nucleic acid and protein synthesis, which may result in cell death (Figure 1I–III) [22]. As to Gram-negative bacteria, AMPs may act as an antagonist on the lipopolysaccharide (LPS) of the outer membrane. LPS is an endotoxin that stimulates the secretion of proinflammatory cytokines to modulate immune responses [13]. This endotoxin is essential in the integrity and stability of bacterial cell structures, providing a protective barrier from chemical attack while managing cell permeability [13,90]. AMPs have demonstrated the ability to inhibit LPS-induced cellular responses via TLR4 and may eliminate extracellular LPS through interactions with this endotoxin in various molecules using membrane permeation methods [13,91]. Such an LPS neutralization usually led to an anti-inflammatory effect.

Amino acids are essential elements that contribute to the antimicrobial activity seen in AMPs. Aspartate and glutamate are negatively charged amino acids responsible for binding divalent cations, which is essential for the antibacterial activity of some AMPs such as daptomycin. When these negatively charged amino acids bind to divalent cations, the conformation of the AMPs is altered to enhance the interaction between the AMPs and cell membranes [92,93]. Tryptophan (Trp) is a non-polar amino acid located on the protein that intersects the membrane interface region [9,94]. The preference of tryptophan residing in the membrane interface is likely the result of tryptophan’s sidechain indole, rigid structure, and aromaticity, which limits access to the hydrocarbon core [95]. Because of its bulky side chain, Trp is frequently used to enhance peptide activity, especially short ones. Arginine and lysine are basic amino acids which contribute to cationic charge, a characteristic seen in amphipathic AMPs [96]. Histidine residues can function as a proton shuttle, which allows these residues to modify the net positive charge by adjusting the pH, affecting antimicrobial activity [97]. When histidine incorporates into the AMP sequence, it can provide a cationic charge at an acidic pH, but it loses a positive charge at a basic pH and reduces peptide activity [96,98]. Cysteine residues can form disulfide bonds, providing stability against chemical, thermal, and enzymatic degradation [93]. Proline residues are frequently found as a structural breaker in an alpha helix due to its properties including steric hinderance and a lack of hydrogen to participate in helix stabilization [99]. Interestingly, there are proline-rich AMPs that inhibit ribosome assembly [100].

The primary antimicrobial mechanisms of AMPs via membrane disruption have demonstrated advantages over conventional antibiotics. AMPs can kill pathogens faster than traditional antibiotics and have a narrower active concentration window to kill pathogens [101]. Typical antibiotics only act on specific intracellular targets, while AMPs utilize various intracellular and extracellular targets. Since AMPs can act rapidly on pathogens and have various killing mechanisms, it reduces the chance of bacteria developing resistance to AMPs [102].

**Figure 1 microorganisms-11-01129-f001:**
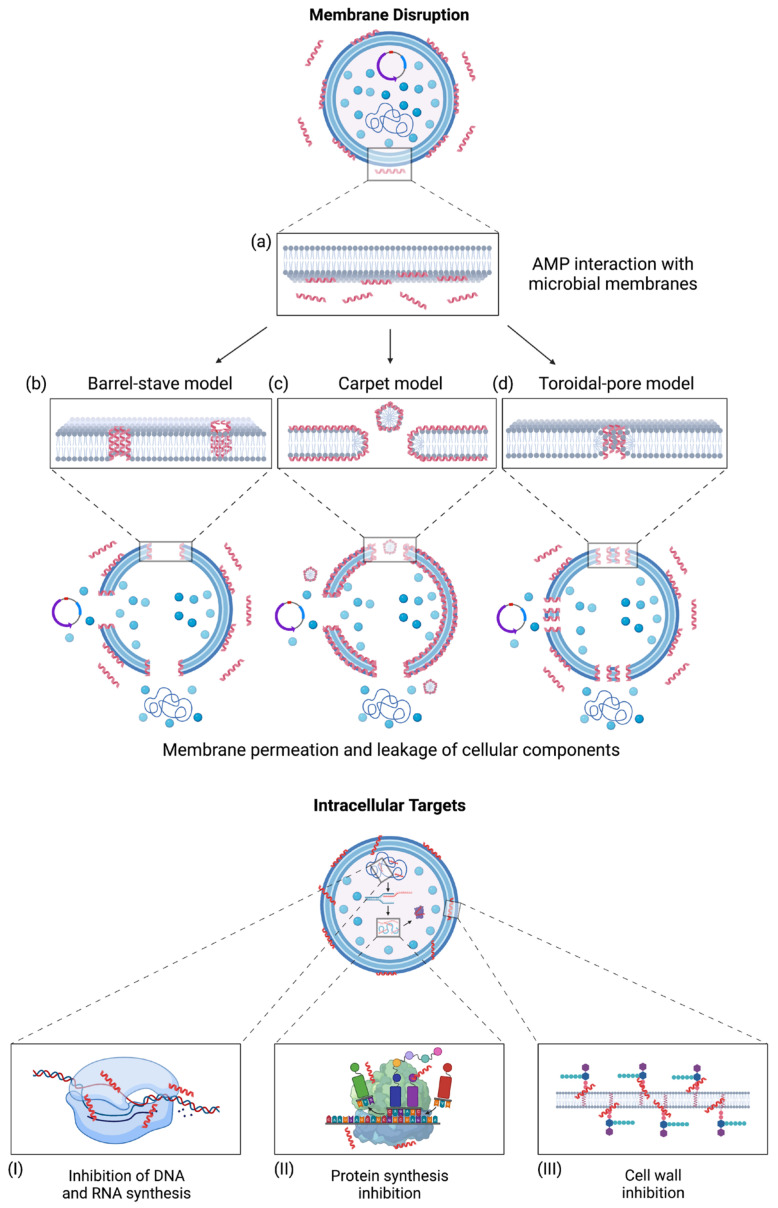
AMPs interact with microbial membranes and their associated mechanisms of action. The mechanisms for membrane permeation begin with (**a**) AMP interaction with microbial membranes, allowing AMPs to insert themselves. (**b**) The barrel-stave model results when AMPs insert perpendicularly into microbial membranes and self-assemble to form pores [83,84]. (**c**) The carpet model works by absorbing AMPs, resulting in disrupting the microbial membranes through micellization [86]. (**d**) The toroidal-pore model results from incorporating AMPs to form a continuous interface with the microbial membrane [28,84]. AMPs can also translocate across microbial membranes and bind to intracellular targets, resulting in cell death. These intracellular targets include (**I**) inhibiting DNA and RNA synthesis, (**II**) inhibiting protein synthesis, and (**III**) inhibition of cell wall synthesis [22]. Adapted from (Bahar et al., 2013) CC BY 3.0 [103]. Created with BioRender.com (accessed on 15 December 2022).

## 4. Limitations and Strategies for Clinical Applications

There is a growing interest in developing biomedical applications for AMPs due to their unique properties and different antimicrobial mechanisms. It appears that the success of AMPs to a large extent is determined by their structure and biological source. Based on a universal peptide classification scheme [24], AMPs are classified into four classes: linear (UCLL), sidechain-connected (UCSS), sidechain-backbone linked (UCSB), and backbone-connected (UCBB). The latter three belong to cyclic peptides. Because of the stability with proteases, some cyclic AMPs have been approved for clinical applications (Table 1). Linear AMPs, however, face challenges due to issues with protease degradation [104].

### 4.1. Clinical Applications of AMPs from Prokaryotes

Currently, there are a few Food and Drug Administration (FDA)-approved peptide antibiotics being used clinically to treat patients. Broadly, they belong to natural AMPs. Table 1 summarizes peptide molecular weight, activity, targeted pathogens, mechanism, administration routes, year of approval, and reference. The structures of these peptide antibiotics are depicted in Figure 2. They can be classified into two types: natural and engineered. Natural peptides include bacitracin [105], daptomycin [106], teicoplanin [107], vancomycin [108], colistin [109], and gramicidin [110]. Bacitracin, daptomycin, teicoplanin, and vancomycin primarily inhibit Gram-positive pathogens, while colistin is the last resort antibiotic to treat Gram-negative pathogens. Gramicidin can kill both Gram-positive and -negative pathogens. Bacitracin is a cyclic heptapeptide with its lysine sidechain attached with a short peptide segment. It is used topically to treat skin infections caused by Gram-positive bacteria, such as *Staphylococcus aureus* (*S. aureus*). Daptomycin and colistin also have cyclic structures attached with a fatty acid tail (e.g., lipopeptides). Vancomycin is a sidechain-linked peptide antibiotic for systemic use. Teicoplanin has a structure similar to vancomycin. It is a mixture of multiple molecules that differ in the fatty acid chain structure. Due to a longer half-life that allows less frequent dosing, teicoplanin has an advantage over vancomycin. Additionally, dalbavancin [111], oritavancin [112], and telavancin [113] are semi-synthetic lipoglycopeptides developed to treat Gram-positive bacterial infections where vancomycin does not work well. They all share a heptapeptide core that inhibits transglycosylation and transpeptidation. Remarkably, dalbavancin and oritavancin have a long half-life in the range of 147 to 393 h, making it possible to administer a treatment of one dose per week [114]. Gramicidin D consists of a mixture of gramicidins (80% A, 6% B, and 14% C). These linear peptides are particularly effective against Gram-positive bacteria by forming an ion channel on membranes. Gramicidin can only be used topically to treat eye, nose, throat, and wound infections since it is highly hemolytic. Note that gramicidin S (AP02243 in the APD database) is very different as it possesses a head-to-tail cyclic backbone structure. Overall, it shows higher activity against *S. aureus* and *Enterococcus faecium* (*E. faecium)* than Gram-negative bacteria [115].

Finally, there are also antiviral peptides (Table 1) engineered based on viral proteins to inhibit viral proteases (e.g., Boceprevir [116] and Telaprevir [117]) or fusion (e.g., Enfuvirtide [118]). Such designed peptides differ from classic AMPs.

**Figure 2 microorganisms-11-01129-f002:**
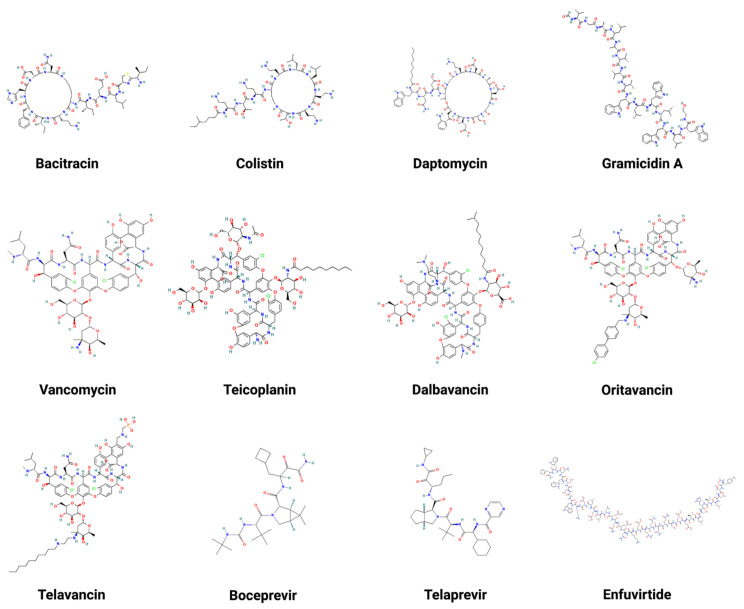
Two-dimensional structures of multiple FDA-approved AMPs. These AMPs include bacitracin [105], colistin [109], daptomycin [106], gramicidin A [110], vancomycin [108], teicoplanin [107], dalbavancin [111], oritavancin [112], telavancin [113], boceprevir [116], telaprevir [117], and enfuvirtide [118]. Created with BioRender.com (accessed on 15 December 2022).

### 4.2. Eukaryotic AMPs under Investigation for Clinical Applications

Multiple AMPs from eukaryotes, such as insects, amphibians, and mammals, have entered clinical trials but have not received FDA approval. For example, omiganan, a derivative of indolicidin, is a recently transitioned Phase III drug that reduces catheter colonization and acts as an anti-inflammatory agent against rosacea, an auto-inflammatory skin disease [13]. Pexiganan, derived from frog magainin, has also been tested to treat diabetic foot ulcers, but has not yet been approved [119]. Its antimicrobial effect is even better when it is used in combination with nisin due to different mechanisms to induce bacterial killing [120]. Iseganan, a derivative of pig cathelicidin protegrin-1, was tested clinically to treat oral mucositis [121].

More AMPs are under development. For example, DP7 (VQWRIRVAVIRK) is an AMP effective against Gram-negative, Gram-positive, and multidrug-resistant bacteria [91,122]. Studies have investigated the effectiveness of DP7 against biofilm production [91] and infections such as the severe acute respiratory syndrome (SARS) coronavirus [123]. A mutation in *Pseudomonas aeruginosa* (*P. aeruginosa)* blocked the generation of a signal molecule, resulting in hindered differentiation and biofilm formation [124]. Yin et al. investigated the antibiofilm activity of DP7 against a mutated strand of *P. aeruginosa*. Increasing the concentration of DP7 showed a reduction in biofilm biomass, which indicated that DP7 might bind to targets involved in biofilm production. This study showed a biofilm reduction in *P. aeruginosa* of 43% to 68% [91]. Zhang et al. investigated the ability of DP7 in resisting the coronavirus infection on the ACE2 receptor by the coronavirus receptor-binding domain (RBD) and spike protein (S protein). The results from their study showed that a 50% inhibitory concentration (IC50) of DP7 inhibiting SARS-CoV and SARS-CoV-2 S protein pseudovirus-infected ACE2-293T was 104 μg/mL and 74 μg/mL, respectively. An enzyme-linked immunosorbent assay (ELISA) was used to determine if DP7 could inhibit SARS-CoV-2 S-RBD from binding to the ACE2, which showed that DP7 inhibited the combination of the two [123]. There are numerous studies aimed at developing human cathelicidin LL-37 into novel antimicrobials (for a review, see ref. [125]).

### 4.3. Complications and Potential Solutions

The use of AMPs for clinical applications, unfortunately, faces limitations. One major issue with many AMPs is their hemolytic activity, which is the ability to burst red blood cells in an organism [126]. This AMP property is a significant concern for therapeutics, as hemolysis may cause anemia or death following treatment. AMPs may also be degraded via proteolytic enzymes found in the GI tract, blood serum proteases, and kidney drug clearance [87,126]. Such degradation results in poor oral bioavailability. Moreover, AMPs are relatively large (on average, there are 33 amino acids in the APD), which can increase their manufacturing costs [13]. Although AMPs may reduce the chance of bacteria developing resistance, there are a few resistance mechanisms reported [102,127]. These resistance mechanisms include modifying cell surfaces to reduce the negative charge on their membranes, resulting in the reduced ability of AMPs to bind to these membranes and kill microbes [128]. Other resistance mechanisms include antimicrobial efflux pumps, external trapping, proteases used to degrade peptides, AMP sequestration to block access to the cell membrane, and biofilms [129,130,131]. There is an increased need for techniques to minimize or remove these limitations associated with AMPs.

New technology and techniques may overcome the challenges facing the applications of AMPs. Nanotechnology, a growing field, has the potential to improve bioavailability, permeability across barriers, protect against harsh environments such as pH and enzymes, and control the release of AMPs [132]. Meanwhile, computational methods can avoid poor peptide candidates initially, as seen with conventional methods of screening peptides using growth media and determining cell toxicity [127]. Rational design, a computer-aided method, of AMPs can reduce the amino acid length of the peptides to perform their desired function against pathogens [133]. This design can reduce the length of AMPs while maintaining their antimicrobial properties, which could help reduce production costs. For example, LL-37 has been reduced to 12 amino acids (KR-12) for peptide engineering, and the recently optimized lipopeptide is composed of only eight amino acids [26,134]. Machine learning models have gained attention due to their cost-effectiveness in peptide discovery [135]. These models can mine through relationships between antimicrobial activity and biochemical features, which help predict AMPs in large-scale environments [136]. Machine learning methods can potentially lead to reduction of toxicity seen in AMPs by modifying the physicochemical features and chemical modifications responsible for toxicity [137].

Although computational methods have helped with obstacles facing AMPs, there are a few computational challenges associated with them. The availability of new sequences such as metagenomes shows their potential to find novel AMP sequences. However, predicting small genes in DNA sequences and AMP activity using homology-based methods is challenging. Due to the limited use of homology-based methods to be directly applied to AMPs, different techniques are required for the application of longer peptides [138]. Santos-Júnior developed Macrel to overcome these obstacles with the use of metagenomes to predict AMP sequences and activity. This method can process metagenomic contigs, metagenomes, or peptides. Macrel can process reads and assemble them into larger contigs, where the small open reading frames (ORFs) are extracted and classified into AMPs or rejected. Since predicting small ORFs with the current methods has issues with false positives, Macrel filters allowed the prediction of high-quality AMP sequences [138]. Huertas-Cepas et al. developed eggNOG-mapper, a tool to annotate large sets of proteins based on orthology assignments from the eggNOG database. This method showed improvements compared to homology-based techniques, revealing that orthologs are effective predictors for AMPs [139].

**Table 1 microorganisms-11-01129-t001:** FDA-approved AMP antibiotics.

Name	MW (g/mol)	Activity	Pathogen	Antimicrobial Mechanism	Administration	Year Approved	Ref.
Bacitracin	1460	Gram-positive	*Streptococcus* spp., *Staphylococcus* spp., *Clostridium* spp., *Corynebacterium* spp., and *Actinomyces* spp.	Bacitracin inhibits the synthesis of bacterial cell walls, which prevents the dephosphorylation of the P-P-phospholipid carrier. This carrier binds peptidoglycan precursors to the bacterial cell membrane. Bacitracin absorbs into injured skin to prevent mucopeptides from entering microbial cell walls.	Topical, ophthalmic, intramuscular	1984	[140,141]
Boceprevir	519.7	Antiviral	Hepatitis C	Boceprevir inhibits the hepatitis C virus (HCV) non-structural protein 3 (NS3/4A) protease, important for cleaving HCV polyprotein into its mature protein forms. HCV replication is inhibited through boceprevir binding covalently to the active site on NS3 protease.	Oral	2011	[142]
Colistin	1750	Gram-negative	*A. baumannii*, *P. aeruginosa*, *E. coli*, *Enterobacter*, *Salmonella*, *Klebsiella*, and *Shigella*	Colistin binds through electrostatic interactions to the negatively charged Gram-negative membrane component, lipid A of LPS. Colistin displacing divalent cations of magnesium and calcium and inserting its hydrophobic acyl fatty chain results in membrane lysis.	Intravenous and intramuscular	1962	[5,143,144]
Dalbavancin	1816.7	Gram-positive	*S. aureus*, *Streptococcus pyogenes*, *Streptococcus dysgalactiae*, *Enterococcus faecalis*, *Streptococcus intermedius*, *Streptococcus agalactiae*, *Streptococcus anginosus*, and *Streptococcus constellatus*	Dalbavancin inhibits cell wall synthesis through interactions with the D-alanyl-D-alanine terminus of the pentapeptide in peptidoglycan of cell wall. This binding prevents crosslinking, which is essential for building the bacterial cell wall.	Intravenous	2014	[5,145]
Daptomycin	1620.7	Gram-positive	*S. aureus*, *Enterococcus faecalis*, *Streptococcus dysgalactiae*, *Streptococcus agalactiae*, *Streptococcus pyogenes*, *Corynebacterium jeikeium*, *Enterococcus faecium*, *S. epidermis*, and *S. haemolyticus*	Daptomycin has a lipophilic acyl tail which inserts into bacterial cytoplasmic membrane, resulting in the efflux of potassium and leading to membrane depolarization. This depolarization results in the inhibition of DNA, RNA, and protein synthesis, resulting in membrane lysis.	Intravenous	2003	[5,146,147]
Enfuvirtide	4492	Antiviral	HIV-1	Enfuvirtide inhibits the fusion of viral and cellular membranes, which prevents HIV-1 from entering cells. Enfuvirtide binds to the first heptad-repeat (HR1) in the gp41 subunit of the viral envelope glycoprotein, preventing confirmational changes necessary for fusing viral membranes.	Subcutaneous	2003	[148]
Gramicidin D	1811.2	Gram-positive	*Streptococcus pneumoniae*, *S. aureus*, *Haemophilus influenzae*, *P. aeruginosa*, *Streptococcus agalactiae*, *E. coli*, *Neisseria meningitidis*, *Klebsiella*, *Neisseria gonorrhoeae*, and *Enterobacter*	In microbial membranes, two gramicidin molecules form a head-to-head dimeric ion channel in the center, resulting in loss of intracellular solutes, reduction in ATP, and inhibition of DNA and RNA synthesis.	Topical	1955	[5,149]
Oritavancin	1989.1	Gram-positive	*S. aureus*, *Streptococcus intermedius*, *Streptococcus pyogenes*, *Streptococcus dysgalactiae*, *Streptococcus agalactiae*, *Enterococcus faecalis Streptococcus anginosus*, and *Streptococcus constellatus*	Oritavancin inhibits biosynthesis of cell walls by binding to the stem peptide of peptidoglycan precursors. Oritavancin may inhibit this biosynthesis by binding to peptide bridging segments. Oritavancin also disrupts the integrity of microbial membranes, resulting in cell death.	Intravenous	2014	[5,150]
Teicoplanin	1900	Gram-positive	*S. aureus*, *S. epidermidis*, *Streptococcus pyogenes*, *Staphylococcus haemolyticus*, *Staphylococcus hominis*, *Staphylococcus saprophyticus*, *Streptococcus bovis*, *Streptococcus pneumoniae*, *Streptococcus agalactiae*, *Streptococcus mitis*, *Streptococcus milled*, *Streptococcus sanguis*, *Clostridium difficile*, *Listeria monocytogenes*, *Clostridium perfringens*, *Propionibacterium acnes*, and *Corynebacterium jeikeium*	Teicoplanin inhibits the peptidoglycan polymerization of the cell wall by binding to Ac_2_-L-Lys-D-ala-D-ala, an analog of the peptidoglycan precursor in cell wall biosynthesis. As a result, teicoplanin inhibits cell wall synthesis.	Intravenous or intramuscular	1990	[151,152]
Telaprevir	679.85	Antiviral	Hepatitis C	Telaprevir inhibits the HCV NS3/4A protease, which is important for cleavage of the HCV polyprotein into mature forms necessary for viral replication.	Oral	2011	[153]
Telavancin	1755.6	Gram-positive	*S. aureus*, *Streptococcus pyogenes*, *Streptococcus constellatus*, *Streptococcus agalactiae*, *Enterococcus faecalis*, *Streptococcus anginosus*, and *Streptococcus intermedius*	Telavancin binds to the D-alanyl-D-alanine terminus of cell wall precursors and late-stage peptidoglycan precursors such as lipid II. This interaction inhibits the polymerization of precursors into peptidoglycan and crosslinking. The mechanism of telavancin and membrane lysis is unknown.	Intravenous	2009	[5,154,155]
Vancomycin	1485.7	Gram-positive	*S. aureus*, *Streptococcus bovis*, *S. epidermidis*, *Streptococcus pyogenes*, *Listeria monocytogenes*, *Streptococcus agalactiae*, *Clostridium* species, *Actinomyces* species, and *Lactobacillus* species.	Vancomycin inhibits the polymerization of peptidoglycans by binding to D-alanyl-D-alanine. This interaction inhibits glucosyltransferase and P-phospholipid carrier, which prevents synthesis and polymerization of the peptidoglycan layer. This inhibition results in a weakened layer and leaks intracellular components.	Intravenous and oral	1954	[5,156,157,158]

Spp = species plural; *S. aureus* = *Staphylococcus aureus*; *P. aeruginosa* = *Pseudomonas aeruginosa*; *E. coli* = *Escherichia coli*.

## 5. Machine Learning for AMP Prediction and Design

Machine learning algorithms are computational techniques that use input data to build an adaptive model that can perform a task without having to be programmed by humans. Due to the adaptiveness of these algorithms, they can modify their architecture through repetition to enhance and adapt their ability to perform a task. This adaptation is known as training, including input data and desired outcomes. Through this training, the algorithm configures itself to produce desired outcomes from not only trained inputs but also unseen data. This training learns outcomes, which is known as machine learning [159].

Machine learning can be divided into categories based on how the data are labeled, including supervised and unsupervised learning. Supervised learning can focus on pattern recognition to distinguish between different data sets based on the data. For example, AMPs and other non-AMPs are unique but still have features that can distinguish one another. Instead of coding every AMP and non-AMP, the program learns to distinguish these through repeated encounters with AMPs and non-AMPs. The input data (cationic, amphiphilic, etc.) pair with the classification label (antimicrobial or non-antimicrobial) to recognize an AMP or a non-AMP (Figure 3a) [159]. Unsupervised learning is different, where this algorithm only uses input data without output data to train the algorithm. The algorithm takes unlabeled data, finds trends or patterns, and learns features from the data. When models encounter new data, the algorithm uses the learned features to recognize and classify the data [160]. For example, peptide databases contain a variety of peptides where the algorithm finds trends in the data to distinguish AMPs from non-AMPs (Figure 3b).

When there is no experimental data, AMP prediction models can be used to find new sequences with AMP characteristics, predict the mechanism or target of AMPs, or estimate three-dimensional (3D) structures. These models can predict AMP features by mining through databases with known AMPs and scanning the literature for sequences with similar features to establish a relationship between the sequences [161]. Peptide databases contain information such as physicochemical features, modifications, mechanisms, and structures to sort the data. Databases can be helpful to researchers when developing peptides through de novo methods and optimizing known AMP sequences. Optimizing AMPs depends on utilizing the physicochemical parameters involved in disrupting microbial membranes [102]. Alpha-helical peptides, for instance, are essential for membrane permeation, where the hydrophobic and hydrophilic faces and net charge influence pore formation [162].

**Figure 3 microorganisms-11-01129-f003:**
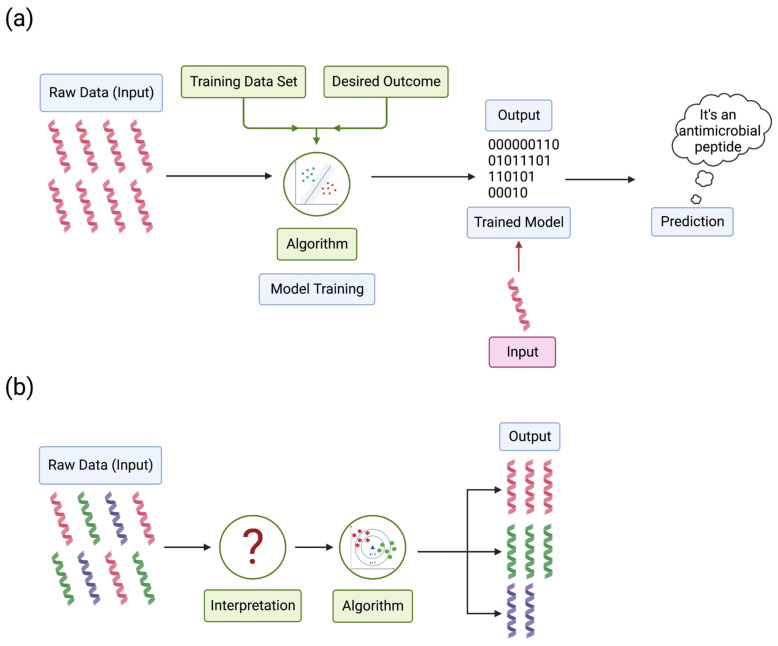
Schematic representation of machine learning algorithms used to develop prediction models. Machine learning algorithms are divided into two main categories. (**a**) Supervised machine learning uses a training set of known AMPs and the desired outcome, such as antimicrobial or non-antimicrobial properties. This model uses raw data of known AMPs to train the algorithm, resulting in a trained model for analyzing new data. Then, when inputting new data into the model, it can predict if that new sequence has antimicrobial or non-antimicrobial properties. (**b**) Unsupervised machine learning does not have a training data set. Instead, unsupervised learning predicts if new sequences are antimicrobial or non-antimicrobial based on any patterns or trends associated with the data. Adapted from (Ma, Y., 2018) CC BY 4.0 [163]. Created with BioRender.com (accessed on 15 December 2022).

### 5.1. Databases for AMP Design

Databases for the machine learning design of AMPs are classified into general and specific databases based on the categories of data collection. General databases (Table 2) provide a broad scope of AMPs, regardless of the source [164]. The AMP database (i.e., APD3), the most studied database, focuses on features such as natural AMPs with less than 200 amino acid residues, including human antimicrobial proteins, known amino acid sequences, and biological activity. This original database, which launched online in 2003, consisted of peptide search, prediction, design, and statistics interfaces [165]. There are currently 3569 AMPs in this database, where users can sort information based on charge, length, and hydrophobicity [19,166]. It contains natural, predicted, and synthetic peptides. This database has taken the lead in annotating the peptide activity (25 types), chemical modifications (27 types), peptide binding targets, and systematic classification of AMPs in numerous ways. The core data set in this database provides a foundation for us to decode the design principles of natural AMPs. The Collection of Antimicrobial Peptides (CAMP) is another database that contains 10,247 antimicrobial sequences [167]. This database includes patents, validated peptides, and predicted antimicrobial sequences based on similarity. CAMP includes sequences, structures, family signatures, AMP activity, source, target organisms, and hemolytic activity. It has also programmed the machine learning prediction interface [167,168]. Linking Antimicrobial Peptides (LAMP2) is another database currently consisting of 23,253 AMP sequences that contain both natural and synthetic AMPs with less than 100 amino acid residues. This database includes AMPs with antibacterial, antiviral, antifungal, antiparasitic, and antitumor activity. LAMP2 also contains primary structure, collection, composition, source, and function details. This database was built by merging data from other databases, such as the APD3 and CAMP [169,170]. Data Repository of Antimicrobial Peptides (DRAMP), another database, provides information about structural, sequence, clinical, physicochemical, and antimicrobial activity [171]. Peptipedia, another database built via data merging, analyzes peptide sequences with the highest number of sequences with biological activity. This database comprises 30 existing databases, including features such as estimating physicochemical and statistical properties. This database contains 92,055 peptides with machine learning techniques to classify the data for sequences [172].

Specific databases (Table 3) focus on the specific types, sources, or characteristics of AMPs instead of a broad overview. These databases can be categorized based on the biological source (plant or animal AMPs), biological activity (antibacterial), 3D structures, or molecular properties [164]. Hemolytik is a specific database with experimental information demonstrating hemolytic activities and their potencies. Since hemolysis is a significant challenge to AMPs being used for therapeutics, using this database may help overcome this clinical obstacle [173]. Thiobase is composed of thiopeptides, which are peptides produced by Gram-positive bacteria. The identification of these peptides depends on the presence of an azole-substituted nitrogen-containing six-membered ring. These peptides have shown potencies for Gram-positive bacteria, including resistant strands, which could help overcome antibacterial resistance [174]. CyBase is a database composed of plant cyclopeptides (i.e., cyclotides), where cyclic proteins derive from ribosome gene products, and cyclization is a post-translational medication [175]. A primary therapeutic obstacle to AMPs is their degradation by proteolytic enzymes; cyclotides are further stabilized by three sets of disulfide bonds have shown resistance to proteolysis [176].

**Table 2 microorganisms-11-01129-t002:** General databases including a broad scope AMPs with no limitations to the source of the data.

Database	Entries	Description	Website	References
APD	3569	This model database currently focuses on natural AMPs (ribosomal or non-ribosomal). First established criteria for data registration. First and detailed classification of over 25 AMP activity (antibacterial, antiviral, hemolytic, anticancer, spermicidal, anti-inflammatory, etc.), peptide source, properties, 3D structure, peptide binding targets, and detailed chemical modifications (>26 types).	https://aps.unmc.edu/database; accessed on 15 December 2022	[19,166]
CAMP	24,243	Experimentally validated, predicted AMP, and patents.	http://www.camp.bicnirrh.res.in/index.php; accessed on 15 December 2022	[167,168]
LAMP	23,253	A database built via database merging.	http://biotechlab.fudan.edu.cn/database/lamp; accessed on 15 December 2022	[169,170]
DRAMP	22,316	Less than 100 residues, mature sequences, AMP activity demonstrated.	http://dramp.cpu-bioinfor.org; accessed on 15 December 2022	[171,177]
dbAMP	26,447	Natural, synthetic, classification based on functional activities.	https://awi.cuhk.edu.cn/dbAMP; accessed on 15 December 2022	[178]
DBAASP	18,878	Synthetic, ribosomal, non-ribosomal, monomers, multimers, multi-peptides.	https://dbaasp.org/home; accessed on 15 December 2022	[179]
MEGARes	~8000	Antimicrobial drugs, biocide, metal, and multi-compound.	http://megares.meglab.org; accessed on 15 December 2022	[180]
ADAM	7007	Natural sources, size, and sequence structure analysis.	http://bioinformatics.cs.ntou.edu.tw/adam/tool.html; accessed on 15 December 2022	[181]

**Table 3 microorganisms-11-01129-t003:** Specific databases provide information on a specific type or source of AMP instead of a broad scope.

Database	Entries	Description	Website	References
Hemolytik	~3000	Hemolytic activity	http://crdd.osdd.net/raghava/hemolytik; accessed on 15 December 2022	[173]
Thiobase	~100	Bacterial thiopeptides	https://bioinfo-mml.sjtu.edu.cn/THIOBASE/index.php; accessed on 15 December 2022	[182]
CyBase	~1771	Cyclic peptides	http://www.cybase.org.au; accessed on 15 December 2022	[175,183]
Defensins Knowledgebase	~360	Defensins	http://defensins.bii.a-star.edu.sg; accessed on 15 December 2022	[184]
Inverpep	774	Experimentally validated, AMPs from invertebrates	https://ciencias.medellin.unal.edu.co/gruposdeinvestigacion/prospeccionydisenobiomoleculas/InverPep/public/home_en; accessed on 15 December 2022	[185]
BACTIBASE	~177	Bacteriocins	http://bactibase.hammamilab.org/main.php; accessed on 15 December 2022	[186]
YADAMP	~2525	Antibacterial peptides	http://yadamp.unisa.it/searchDatabase.aspx; accessed on 15 December 2022	[187]
Peptaibol	~317	Peptaibol	http://peptaibol.cryst.bbk.ac.uk/home.shtml; accessed on 15 December 2022	[188]
DADP	~2571	Anuran defense peptide	http://split4.pmfst.hr/dadp; accessed on 15 December 2022	[189]
BaAMPs	~221	Antibiofilm peptides	http://baamps.it; accessed on 15 December 2022	[190]
CancerPPD	~3492	Anticancer peptides	http://crdd.osdd.net/raghava/cancerppd; accessed on 15 December 2022	[191]
ParaPep	~863	Antiparasitic peptides	https://webs.iiitd.edu.in/raghava/parapep/peptide.php; accessed on 15 December 2022	[192]

### 5.2. Machine Learning Classifications

Machine learning adopts computation approaches to predict the structure and activity of AMPs through the classification of features into peptide-level and amino-acid-level features. The peptide-level features include sequence-based and structure-based features [193]. These computational approaches use statistics to sort and learn patterns in large amounts of data. Machine learning tools use the data provided by a database, known as a training set, to allow the models to learn and select the best features for the desired outcome. Validating this hypothesis on the new data uses a test set to ensure this model is effective [102]. Machine learning or deep learning models are used in the prediction and generation of AMPs. This scheme with input features, the construction of models, and prediction is shown in Figure 4.

#### 5.2.1. Sequence-Based Features

Sequence-based features are dependent on the composition of amino acids or groups of amino acids to compute vectors [193]. To compute these vectors, various techniques are used, including one-hot encoding, amino acid composition, pseudo-amino acid composition, reduced amino acids, physicochemical properties, etc. [194]. One-hot encoding is a technique that represents a value as a vector of bits, where the vector length is proportional to the number of categories. In the case of AMPs, there are 20 natural amino acids which are each assigned to a bit position. The assigned bit is identified with a value of 1, while the other positions are set to 0. According to this positioning system, alanine (A) is the first amino acid alphabetically and would be positioned at the rightmost bit (00000000000000000001), while tyrosine (Y) would be assigned to the leftmost bit (10000000000000000000) [195]. Amino acid composition is a technique where the vector has a proportion of an amino acid relative to the sequence length. For AMPs, these peptides can be divided into sections with both an N and C terminus, which is used to calculate local amino acid composition. Unfortunately, the amino acid composition does not account for the sequence order effect, which can have a wide range of combinations depending on the sequence length. The pseudo-amino acid composition accounts for the sequence order by computing the correlations and occurrence number between various clusters among a pair of amino acids, resulting in a dimensional vector [193,194]. Although amino acid and pseudo-amino acid compositions have their advantages, they do not account for the specific amino acid sequence for a peptide due to limitations associated with the sequence order and sequence variations [194]. Reduced amino acid compositions group similar amino acids based on sequence correlation factors, such as physicochemical or structural properties. Physicochemical properties include hydrophobicity, charge, and a few other features that are determined through wet lab experiments. The use of physicochemical features are usually in combination with other techniques, such as pseudo-amino acid combinations, to enhance the accuracy of the techniques [194].

#### 5.2.2. Structure-Based Features

Structure-based features look at the secondary structure of proteins or peptides, which has an impact on the antimicrobial activity of peptides. Therefore, predicting AMPs using a combination of sequence-based and structure-based features is essential. Structure-based features are computed using quantitative structure–activity relationship (QSAR), distance distribution, general structure encodings, and other techniques. QSAR is a technique used to describe amino acid sequences through chemical properties [193,194]. This technique can predict antibacterial activity by relating descriptors with peptide properties such as toxicity [196]. The structure of the peptide helps identify descriptors, which include amphiphilicity, net charge, and length. These descriptors help define classifiers and are used to classify the peptides based on their efficacy [14]. Distance distribution involves prediction based on the distribution of distances for each pair of atom types. This technique computes the distance between atom types (such as donor–donor, donor–acceptor, etc.) into a distribution function [197]. General structural encodings differ from QSAR methods in that this technique provides structure information from the whole peptide and converts this information to a numerical representation. This technique describes the peptide structure by the 3D shape and is based on the electrostatic potential. General structure encodings can be applied to predicting AMPs because the 3D composition of peptides influences the antimicrobial activity. Overall, the combination of sequence-based and structure-based features enhances the accuracy and effectiveness of the prediction [194].

#### 5.2.3. Amino-Acid-Level Features

Amino-acid-level features look at the sequence, where each word corresponds to the one-letter code of the amino acids. These features are primarily used in deep learning algorithms, but can use embedding layers to extract representative features such as word embedding and contextualized embedding [193]. Word embedding techniques involve Word2vec and Global Vectors (GloVe). Word2vec is a shallow word embedding model that predicts words based on their context using neural models such as a continuous bag of words (CBOW) or Skip-Gram. CBOW uses three layers to predict a word, where the input layer describes the context. The hidden layer describes the projection of the words from the input and is projected into the output layer. Skip-Gram is different from CBOW in that the input layer relates to the target word, while the output layer relates to the context. Skip-Gram predicts the context of the word as opposed to the prediction of a word. GloVe is a technique based on the occurrence of words in a textual corpus. This technique involves constructing a co-occurrence matrix and factorization of the matrix to obtain vectors [198]. Contextual embedding is different from word embedding in that the embedding learned is a function of the whole input sequence, therefore allowing the same word to have various representations in different contexts. Contextual embeddings are generated through deep learning techniques and use language models, such as bidirectional encoder representation from transformers (BERT), text-to-text transfer transformer (T5), and auto-regressive model (XLNet) [193]. BERT is a technique used to pre-train deep bidirectional representations using unlabeled text, where context is conditioned in the layers. This technique is able to pre-train bidirectional representations by masking a percentage of input tokens randomly and predict those tokens through a masked language model (MLM). The pre-trained deep bidirectional representation allows the model to be fine-tuned with an additional output layer to provide a range of tasks [199]. T5 considers a task and is inputted into the model as text and trained to generate target text [200]. XLNet models extract bidirectional contextual information through the random ordering of the input order while maintaining the one-way model. This bidirectional contextual information is attained by considering the ranking order of the text, but it only predicts a partial sequence [201].

#### 5.2.4. Machine Learning Models

Machine learning uses models or algorithms to classify data and predict an outcome based on the provided data. To predict antimicrobial activity, these models use statistics to learn the relationships between physicochemical properties and amino acid composition. Various algorithms predict the AMP activity, mechanism of action, and efficacy. Prediction models for AMPs include k-nearest neighbor (KNN), support vector machine (SVM), artificial neural network (ANN), random forest (RF), and other algorithms. These models can be trained by using features, such as amino acid composition, length, molecular weight, charge, and hydrophobicity [102].

KNN (Figure 5a) performs a classification and regression analysis of data points to find a trend. Classifying data records (t) results from retrieving its k nearest neighbors, forming a neighborhood of t [202,203]. The nearest neighbor is the data point closest to the desired data point. When the nearest items are determined, the algorithm recommends these items to the users based on the previous majority of nearest neighbors [204]. This model is dependent on selecting a k value, where the effectiveness of the classification is dependent on this value. Due to this dependence, this model is limited based on selecting the appropriate number of neighbors for the algorithm [202].

A supervised machine learning model known as SVM (Figure 5b) classifies data by finding a hyperplane to segment collected data. This model creates a hyperplane with the largest distance from the nearest training points. The maximal distance created by the hyperplane allows a strong separation between classes, resulting in a lowered generalization error [203,205]. SVM can perform a linear classification through regression analysis and non-linear classification using the kernel trick. The kernel trick works by mapping data into high-dimensional feature spaces, characterizing data in multiple dimensions [160]. Unfortunately, limitations with this classifier result from data in target classes overlapping, resulting in an uncertain classification of data [203].

ANN (Figure 5c) is a supervised machine learning model that uses various hyper-parameters to approximate a relationship between input and output values. These hyper-parameters include many hidden layers and units, a learning rate, and activation function [203]. ANN is composed of computational units called neurons, which are organized into layers and connected to form a network. Artificial neurons can receive signals from various sources and transform them through an activation function [206]. The general structure of an ANN consists of input, hidden, and output layers. The input layer receives incoming data, hidden layers transport incoming data into a higher-order function, and an output layer makes a prediction about the input data. Having multiple hidden layers helps obtain high non-linearity, preventing the computation of a large linear function [203,207].

By building multiple decision trees, random forests (Figure 5d) can overcome obstacles associated with decision trees, such as overfitting data due to outliers. Decision trees are algorithms used to classify an event or predict the outcomes of a variable. Decision trees primarily solve classification problems but can also perform regression analysis. Random forests function through the random sampling of the data while building several decision trees for each random sample of data [204]. Building several decision trees with controlled variation requires a combination of bootstrap aggregation and random feature selection. Bootstrap aggregation (bagging) is a method that generates and averages predictors when forming outcomes by using a majority vote to predict a class [203,208].

**Figure 5 microorganisms-11-01129-f005:**
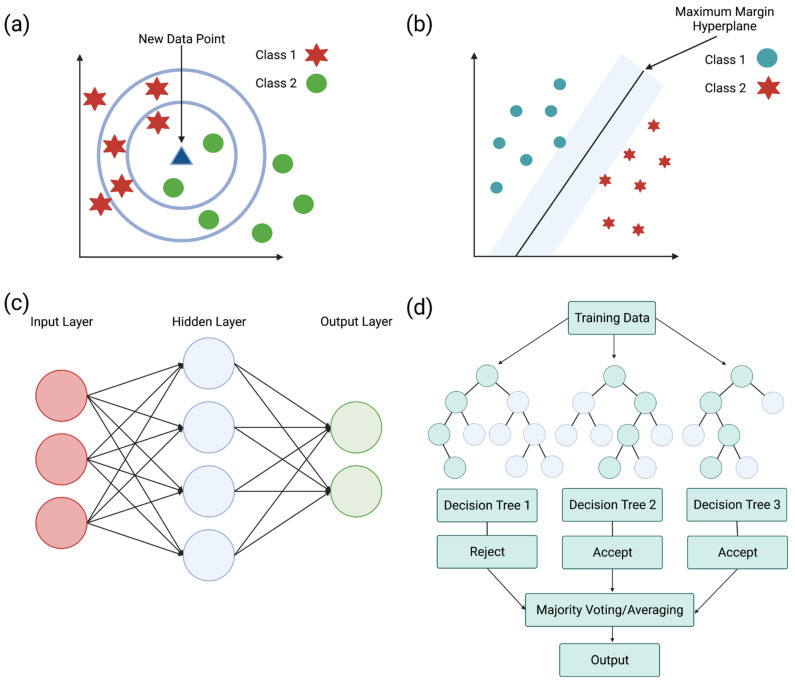
Machine learning uses various models to classify and predict outcomes based on data. (**a**) k-nearest neighbor (KNN) classifies data using the nearest neighbor, which is the data point closest to the desired point. When the nearest items are determined, the algorithm recommends using these items based on the previous majority vote of nearest neighbors [204]. (**b**) Support vector machine (SVM) is a machine learning algorithm that classifies data by finding a hyperplane to separate the data. A hyperplane creates a line that is the maximum distance from the nearest training points, resulting in a strong separation between classes [203,205]. (**c**) Artificial neural network (ANN) uses hyper-parameters to approximate relationships between input and output values. ANNs are composed of computational units, which receive signals and transform them to make a prediction using the network formed by layers [203,206]. (**d**) Random forests build multiple decision trees to classify an event and avoid issues such as overfitting data due to outliers [204]. Adapted from CC BY-SA 3.0 [209], CC BY 4.0 [210], CC BY-SA 3.0 [211], and CC BY-SA 4.0 [212].

### 5.3. Machine Learning Prediction and Design of AMPs

As discussed earlier, machine learning is a time-saving technique that can perform a task, such as predicting an AMP, without being programed by humans, and can modify the algorithms to adapt to the task [159]. These techniques are applied to the prediction of AMP sequences to obtain unique properties to avoid potential issues with AMPs. For instance, machine learning can be applied to develop non-hemolytic AMPs, which is a limitation in their therapeutic use. Overall, with increasing knowledge on machine learning, this technique can be used to mitigate the issues faced in wet lab experiments.

#### 5.3.1. Predicting AMP Sequences

Machine learning methods have been used to predict AMPs soon after the establishment of the original antimicrobial peptide database in 2003, known as the Antimicrobial Peptide Database (APD) [165]. Lata et al. developed the first SVM model, named AntiBP2, to predict AMPs using the data collected from the APD. This SVM model is based on the amino acid composition of the peptide using a five-fold cross-validation technique and achieved a 92.14% accuracy [213]. Additionally, based on the APD, Xiao et al. developed a fuzzy K-nearest neighbor algorithm based on pseudo-amino acid composition. This algorithm is a multi-label classifier where the pseudo-amino acid composition components incorporate various physicochemical properties. This method was named iAMP-2L and achieved an accuracy of 86.32% when identifying AMPs and non-AMPs [214]. Torrent et al. created an ANN method that correlates physicochemical properties with antimicrobial activity. This model used the CAMP peptide database for the positive data set and the Uniprot database for the negative data set with an overall accuracy of 90% [215]. Joseph et al. developed a one-against-all classifier model using random forests and SVMs to predict sequences as antimicrobial or non-antimicrobial. In this multi-classification model named ClassAMP, random forests determined the essential features for classification. ClassAMP achieved an overall accuracy of about 95% with the test sequences [216]. Lawrence et al. developed a random forest classifier named amPEPpy to predict AMP sequences. The number of used trees in amPEPpy was optimized using the out-of-bag (OOB) error, where 128 decision trees were optimal with an OOB error of 0.036 [217].

#### 5.3.2. Machine Learning for Peptide Design

Recently, Capecchi et al. developed a recurrent neural network (RNN) method to predict antimicrobial and hemolytic activity [218]. RNN is a branch of neural networks that incorporates feedback connections from the previous time step in the model [219]. The preference for using RNN over other models, such as SVM, random forest, and naive Bayes, is due to its better prediction of antimicrobial and hemolytic activity. The RNN model obtained an accuracy of 76% on hemolysis prediction and antimicrobial activity [218]. Sharma et al. developed a web server to predict antibiofilm peptides known as dPABBs, which used amino acid composition and selected residue features using six SVM and Weka models. dPABBs was able to generate an accuracy, sensitivity, and specificity of 95.24%, 92.50%, and 97.73%, respectively, based on the training data sets [220]. Zhang et al. designed a 12-amino-acid-long AMP called DP7 using machine learning methods based on amino acid activity. DP7 has demonstrated broad-spectrum antimicrobial activity in vitro and in vivo. The results from this study confirmed that DP7 could reduce methicillin-resistant *Staphylococcus aureus* (MRSA) bloodstream infection in mice up to day 7 post-infection at doses of 0.5, 1, and 2 mg/kg with 70, 80, and 90% reductions, respectively. DP7 also showed its effectiveness compared to current antibiotics, where the protection level of DP7 at 1 mg/kg was equivalent to 10 mg/kg of vancomycin [221]. The lower concentration indicates that DP7 is more potent than vancomycin at treating the resistant bacteria.

## 6. Peptide Engineering

The proteases produced by human cells and microbes may limit the bioavailability of AMPs. For instance, LL-37 can be degraded by *S. aureus* proteases, aureolysin, and the V8 protease, thereby losing its antimicrobial activity [222]. To avoid the potential inactivation of AMPs, peptide engineering techniques have been studied, including peptide backbone modifications, cyclization, terminal modifications, and substituting L-amino acids with D-amino acids or unnatural amino acids [223]. The most common terminal modifications include N-terminal acetylation and C-terminal amidation, which provide the peptides with different functions. N-terminal acetylation has shown to result in an increased stability and helical content, which can contribute to a deeper insertion into the hydrophobic region in microbial membranes [224]. C-terminal amidation has shown to enhance antimicrobial activity and reduce hemolytic properties, which are advantageous features for antimicrobial peptides [225]. Li et al. computationally designed antimicrobial peptide L163, which is effective against multidrug-resistant bacteria but is degraded by proteases. The results showed that the N-terminal acetylation of L163 had increased stability and reduced host toxicity [226]. Due to human proteases solely recognizing L-amino acids to be degraded, D-amino acids have shown enhanced stability against protease degradation [227]. Lu et al. synthesized derivatives of AMP Pep05 (KRLFKKLLKYLRKF) through the substitution of D-amino acids and unnatural amino acids. The results showed that substitutions with D-amino acids and unnatural amino acids enhanced the ability of peptides to resist proteolytic cleavage from proteases produced by *S. aureus* and *Escherichia coli* (*E. coli)*. When all the L-amino acids of Pep05 were replaced with D-amino acids, this resulted in the highest stability against proteases, but exhibited severe toxicity in vivo [228]. Wang et al. found that the partial incorporation of D-amino acids and biphenylalanines into the major antimicrobial peptide of LL-37 led to selective, stable, and potent antimicrobials against antibiotic-resistant bacteria, including MRSA [229]. An eight-amino-acid lipopeptide of LL-37 made in D-amino acids, which remained stable to five proteases, showed in vivo efficacy against MRSA [134]. More recently, White et al. developed CD4-PP, a synthetic peptide, through the dimerization and head-tail cyclization of the shortest antimicrobial region of LL-37. This cyclized form was stable against aureolysin for 6 h in comparison to LL-37, which degraded within minutes [230]. These results indicate the importance of modifying peptides to enhance the stability required for clinical applications.

## 7. Nanotechnologies for AMP Delivery

Nanotechnology is a growing field involved in applying structures and systems at the nanometric scale, but it has now been implemented into the delivery of drugs with a limited therapeutic efficacy [231,232]. Nanotechnology can also protect AMPs from degradation and increase their efficacy [233]. Nanoparticles range from 0.1 to 100 nm and can transport peptides across the intestinal barrier to enter the bloodstream [132,233]. Various nanomaterials used for the delivery of AMPs include metal nanoparticles (gold, silver), lipid nanoparticles (liposome), polymer nanoparticles [chitosan, hyaluronic acid, and poly(glycolide-co-lactide) or PLGA], and other nanostructures (dendrimer, carbon nanotube, and quantum dot), as shown in Figure 6 [234].

### 7.1. Metal Nanoparticles

Gold nanoparticles do not have antimicrobial activity, but they function as delivery devices due to their ability to enhance the antimicrobial activity of AMPs. Gold nanoparticles may lead to increased antimicrobial activity due to the penetration of bacterial cell membranes and a high surface-to-volume ratio for drug delivery [235]. Rai et al. developed an AMP-conjugated nanoparticle with a gold core and a hydrophilic cationic peptide shell. This gold nanoparticle had a high concentration of AMPs with more significant antimicrobial activity in serum and in proteolytic environments compared to soluble AMPs [239]. Silver nanoparticles may deliver silver ions into bacterial cytoplasm and membrane and present antimicrobial properties [240]. Mei et al. synthesized silver nanoparticles conjugated with AMPs (bacitracin A and polymyxin E). The minimum inhibitory concentration (MIC) of silver nanoparticles conjugated with these AMPs was found to be significantly lower than silver nanoparticles without AMPs [241].

### 7.2. Lipid Nanoparticles

Liposomes are more biocompatible than metal nanoparticles because their composition is similar to cell membranes [242]. The ability of these carriers to be biodegradable and non-hemolytic gives them an advantage over other nanocarriers [236]. Li et al. developed a liposome delivery system loaded with daptomycin and clarithromycin to treat MRSA infections. This antibiotic delivery system had a reduced mutant prevention concentration and decreased mutant selection window, demonstrating its potential to reduce MRSA infections [243]. Sharaf et al. designed liposomes loaded with clarithromycin and bioflavonoid Hesperidin to provide a controlled drug release to target bacterial cell membranes. These liposome nanostructures effectively inhibited *Helicobacter pylori* growth with no cytotoxic effects (IC_50_ < 50 µM). These liposome carriers were biocompatible, which is essential for effectively and safely delivering drugs to their target [244]. Solid lipid nanoparticles (SLNs) were also studied due to their high drug loading and small size [245]. Severino et al. developed an SLN loaded with polymyxin B to treat bacterial infections. This polymyxin-B-loaded SLN was effective against various resistant strains of *Pseudomonas aeruginosa* (*P. aeruginosa*) [246].

### 7.3. Polymer Nanoparticles

PLGA nanoparticles have been well studied for drug delivery. PLGA nanoparticles have low systemic toxicity due to their ability to undergo hydrolysis, producing biodegradable monomers (lactic acid and glycolic acid) [247]. Cruz et al. developed PLGA nanoparticles loaded with an AMP, GIBIM-P5S9K, to determine the antimicrobial activity against *P. aeruginosa* and MRSA. These AMP-loaded nanoparticles showed lower hemolytic activity and higher antibacterial activity against bacterial growth at a lower concentration than free AMPs [248]. Among the polymer nanoparticles studied, chitosan is a biodegradable polymer with broad-spectrum antimicrobial activity and has been generally recognized as safe (GRAS) by the FDA. Due to these properties and being inexpensive, chitosan has been used in therapeutic applications [237,249]. Rashki et al. developed chitosan nanoparticles (CS-NP) loaded with LL-37 to improve its antibacterial and antibiofilm activity. These nanoparticles showed a 6-log reduction in colony-forming units of MRSA compared to free LL-37. The CS-NP loaded with LL-37 also inhibited MRSA biofilm formation to 68% after 2 h compared to 74% for free LL-37 [250]. Hyaluronic acid nanoparticles are another example. Hyaluronic acid is a component in the extracellular matrix that provides biocompatibility for therapeutic applications [251]. Lequeux et al. created an antimicrobial hydrogel by covalently binding nisin to hyaluronic acid. This hydrogel demonstrated an antibacterial efficacy of 99.99% for *Staphylococcus epidermidis* and 99.95% for *S. aureus* at a polymer concentration of 2 mg/mL and a nisin concentration of 0.01 mg/mL [252].

### 7.4. Other Nanostructures

Dendrimers are hyper-branched polymer molecules that have demonstrated effectiveness against biofilms, multidrug-resistant bacteria, and viruses [238]. Jiang et al. developed a PAMAN dendrimer conjugated with vancomycin and silver nanoparticles to kill vancomycin-resistant *Staphylococci*. This dendrimer demonstrated the ability to kill vancomycin-resistant strains of *S. aureus* with an MIC of 2 μg/mL compared to 8 μg/mL for free vancomycin [253]. Carbon nanotubes were also studied, and they can provide antimicrobial activity by oxidizing glutathione, which increases oxidative stress on microbes [254]. Qu et al. developed multi-walled carbon nanotubes, where nisin was covalently immobilized. These carbon nanotubes reduced biofilm formation by 100-fold on deposit film and 2.6-fold in the suspension. The antimicrobial activity of nisin had retained at least 90% of bacterial activity for carbon nanotubes with immobilized nisin [255]. More recently, quantum dots were studied. Quantum dots have a high transfer of electrons, allowing them to produce free radicals such as reactive oxygen species (ROS), resulting in antimicrobial activity and microbial cell death [256]. Zhao et al. developed nitrogen-doped carbon quantum dots (NCQDs) to assess their antimicrobial activity against antibiotic-resistant bacteria. These NCQDs had an MIC of 0.128 mg/mL and 0.256 mg/mL for methicillin-resistant and susceptible *S. aureus*, respectively, with no toxicity to the host’s organs [257].

## 8. Summary and Perspective

Antibiotic resistance is a significant complication in healthcare since doctors may face situations where they have no antibiotics to use. AMPs have been shown to present a broad range of protection against pathogenic microorganisms in vitro, and more are under investigation for their use as alternatives to antibiotics. Until recently, researchers discovered new AMPs by modifying known AMPs to be individually tested against various bacteria to determine antimicrobial efficacy in wet laboratories [127]. These methods are time-consuming, costly, and require many resources [258]. This article has described the use of machine learning methods as an alternative method to identify AMPs, which is anticipated to accelerate the discovery of novel antimicrobials. AMP activity has been predicted based on 2D features, such as amino acid sequence, net charge, and hydrophobicity. Machine learning could be applied not only to predict AMP sequences but also to predict the 3D structure of AMPs. Predicting the 3D structure of AMPs through machine learning would be a helpful tool for researchers to understand the binding of molecules to AMPs, which plays a major role in drug discovery. More recently, machine learning has also been used to predict AMP sequences to target SARS-CoV-2. Using in silico models, Liscano et al. determined two AMPs (caerin 1.6 and caerin 1.10) which showed a potential to interact with the spike surface viral protein (SGP) on SARS-CoV-2 as opposed to the ACE2 protein. The SGP protein is located on the envelope protein of SARS-CoV-2 that plays a role in the binding and fusion of and entering host cells. The results from these in silico experiments show that these AMPs have the potential to block the S protein and ACE2 during viral binding and entry, but need to be experimentally validated to determine their effectiveness [259]. Nanotechnology with antimicrobial properties also has the potential to improve oral health concerns, such as endodontic infections. Currently, there are no available techniques that remove biofilm while not affecting the root dentin [260]. Nanomaterials provide unique properties, including the removal of biofilms; they also avoid demineralization and stimulate re-mineralization [261]. With the limited therapeutic techniques to treat endodontic infections, nanocarriers may be a promising approach to combat oral health concerns.

Therefore, computational technologies have great potential in designing future antibiotics. In this regard, the database filtering technology has illustrated the in silico design of novel antimicrobials with potent activity both in vitro and in vivo [262,263]. Likewise, with the accumulation of data in databases (Figure 7), machine learning algorithms may be assembled into a pipeline to predict sequences to overcome the issues faced with AMPs [19]. These obstacles include degradation by enzymes, poor oral bioavailability, and hemolytic activity. As an example, Plisson [135] et al. developed models to predict hemolytic activity, which predicted that 30% of AMPs were non-hemolytic, where 91% of the predictions were considered reliable. Based on these models, designing non-hemolytic peptides should include neutral or slightly charged sequences with an equal amount of aromatic/aliphatic residues and small amino acids. These machine learning models provided information on non-hemolytic AMP sequences, which can help researchers develop AMPs with minimal toxic effects. Together with other techniques such as peptide engineering, formulations, and nanotechnology, researchers may identify novel AMPs with improved properties and bioavailability for future applications [264].

In summary, this review focuses on nanotechnology approaches for AMP delivery, the roles of AMPs in various diseases, and recent advances in AMP design via machine learning.

**Figure 7 microorganisms-11-01129-f007:**
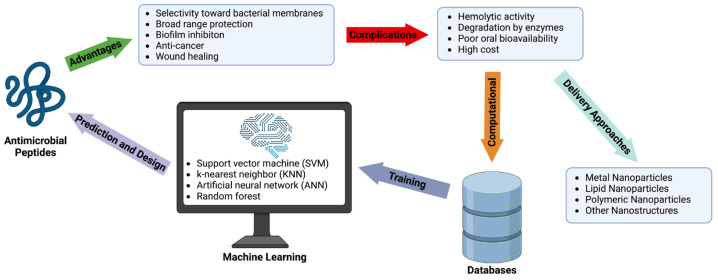
Overall scheme of the advantages of AMPs and different approaches to improve challenges associated with therapeutic applications. AMPs have advantages over current treatments due to their broad range protection and other properties not found in antibiotics. However, these peptides have obstacles when being applied to therapeutic use due to complications such as toxicity and instability. Delivery approaches are used to overcome some of these issues with AMPs. Machine learning has shown to be a promising approach for using models to predict and design AMPs with less therapeutic complications. Databases help train machine learning models to enhance prediction methods for AMPs. Created with BioRender.com (accessed on 21 December 2022).

## Figures and Tables

**Figure 4 microorganisms-11-01129-f004:**
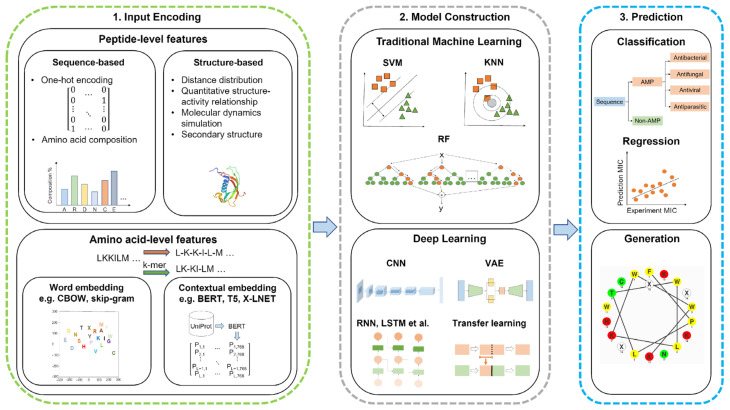
A general machine learning workflow of AMP discovery and design, including a summary of the major techniques in each stage of the workflow. Redistributed from (Yan et al., 2022) CC BY 4.0 [193].

**Figure 6 microorganisms-11-01129-f006:**
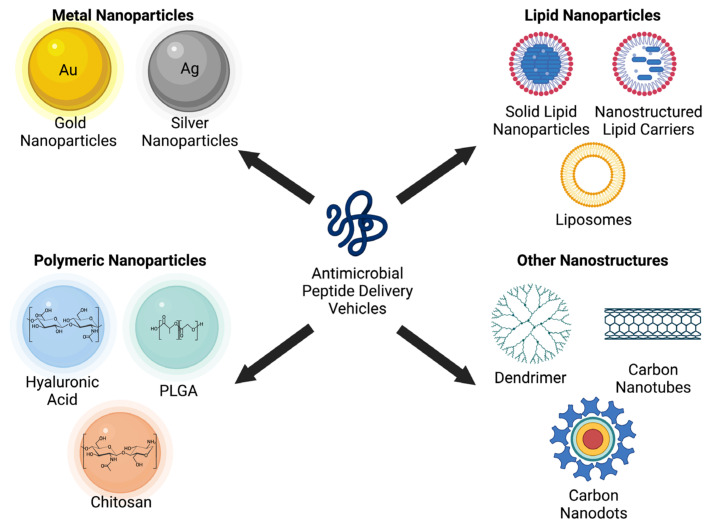
Nanoparticle delivery vehicles for AMPs to enhance biological stability. These delivery vehicles have different purposes based on the properties desired for delivery. Metal nanoparticles have demonstrated the ability to enhance the antimicrobial activity of AMPs but have limitations at high concentrations due to potential toxicity [235]. Lipid nanoparticles are biocompatible, which gives them the advantage of being used to avoid toxicity in tissues [236]. Polymeric nanoparticles contain a coating of molecules with specific properties such as antimicrobial activity and good bioavailability [237]. Forming other nanostructures helps combat complications, such as different structures, allowing biofilm inhibition [238]. Modified with permission from CC-BY 4.0 [235]. Created with BioRender.com (accessed on 21 December 2022).

## Data Availability

No new data were created in this study. Data sharing is not applicable to this article.

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
