# Peer review of "Advances in Antimicrobial Peptide Discovery via Machine Learning and Delivery via Nanotechnology"

_microorganisms, 2023, doi:10.3390/microorganisms11051129_

Round 1
Reviewer 1 Report
Manuscript entitle “Advances in Antimicrobial Peptide Discovery via Machine Learning and Delivery via Nanotechnology” submitted by Sowers et al gathered relevant information about the actual state of the developing AMPs through technology and the possibilities for it delivering. Authors emphasized the relevance of AMPs in some diseases with special relevance of LL-37.
Some AMPs have also shown anticancer activity, this is the case of the work published by Gaspar et al (2015) where the antimicrobial peptide, HNP1 was use as anticancer peptide and I think that this work could enriched this review in section 2.3.
Between the autoimmune diseases we can find type 1 and 2 diabetes, which is associated with reduced AMP expression in the gastrointestinal tract, contributing towards epithelial barrier dysfunction contributing to autoimmunity in the pancreas (doi.org/10.1053/j.gastro.2021.12.272 and doi.org/10.3390/nu15030754). I think a small discussion could be added in section 2.2 about these chronic disease.
After revision and with minor adjustments to the manuscript I consider that authors compiled relevant research and can be accepted for publication.
Reviewer 2 Report
The aim of this review was to focus on the nanotechnology approaches for AMP delivery and advances in AMP design via machine learning. This research is under the scope of this journal; the topic is relevant for readers, and this research deals with potentially significant knowledge of the field. And It will be necessary for Microorganisms knowledge. The topic is relevant for readers and this study deals with potentially significant knowledge in the field and opens new ways for future studies.
However, there are Major aspects which are needed to be improved in the manuscript:
(Abstract)
- please improve the abstract, based on the instructions for the authors. The authors need to reformulate completely the abstract and add more information to the description of the search methodology used and the Results.
(Keywords)
- Please add more keywords, and order the keywords / Mesh terms alphabetically
(Introduction)
What is the importance of this study? What is the gap in this field of literature?
- You do not think this study is included in the others already done? Which results are comparable with other studies? What has this study been new? And Compared with these reviews, Antimicrobial peptides (AMPs), such as LL37 peptides, may be immobilized on the surface of medical devices to render them with antimicrobial and angiogenic properties. please read Akhilesh Rai et al. 2021, 2022 https://doi.org/10.1039/D1BM01034D; https://doi.org/10.1039/D1TB02617H.
(Results)
- Improve the resolution quality of all figures and graphs (and a presentation). The font/language in the figure/caption differs from the text. Please, standardise the size and the font in the figures and charts with the font of the manuscript.
- This section would better communicate to readers if restructured. A flowchart or diagram of the article selection would be valuable.
(Discussion)
- Please, identified what was the strength(s) of this study. And read https://doi.org/10.1016/j.addr.2023.114731 for the implications for future perspectives in oral health.
References
- The titles of references have a different formats,
the title of the article is written in capital letters at the beginning of words, others only in lowercase. Also, the standardized format of the presentation is in the journal's name. Because names have been written in different formats, one is not abbreviated, and the others are not.
Reviewer 3 Report
Line 15: Why?
Line 30: I would explain why this sentence was written and add a reference.
Lines 33-34: Add reference for mechanism of resistance.
Line 37: Only in the United States, and in the other countries? Obviously by adding world information, you have to change the citation.
Line 42: Add references (for example doi: 10.3389/fcimb.2021.668632).
Line 45: Add reference.
Lines 52: Add references.
Line 54-55: Add references (for example https://doi.org/10.1007/s00018-021-03784-z, doi: 10.2174/0929867325666180713125314).
Lines 56-58: Why does the presence of these amino acids influence the peptide charge? Add reference.
Lines 62-64: I would elaborate this sentence or remove it. Inserted like this it makes no sense.
Lines 70-71: The peptide selectivity is caused by the favored interaction of cationic AMPs with anionic bacterial membranes.
Line 78: Add reference.
Lines 82-83: Add references.
Line 86: Add reference.
Lines 90-91: add reference.
Line 94: Add reference.
Lines 96-107: I would rearrange this paragraph, from "chronic obstructive" in 96 lines on. I would divide the description into situations when you have overexpression and when you have a down-regulation for clarity of reading.
Lines 109-113: Add references.
Line 141-142: Add reference.
Lines 142-143: "Due to dysregulation" about what?
Line 143: “Influences cancer biology” By which mechanism? Add reference.
Line 149: Add general reference, after the example reference (already added in [36])
Line 154: Add reference.
Line 158: Add general reference.
Line 170: Add reference.
Line 177: “AMPs” like... add the examples.
Line 178: Add reference.
Line 180: It is redundant because cardiovascular diseases include in the name the conditions involving the heart or blood vessels.
Line 182: Add reference.
Line 190: Add reference.
Lines 199-200: Add references.
Lines 202-203: This sentence is not understood because "including Alzheimer's and Parkinson's" refers to "diseased areas" but they are a disease, not diseased areas.
Line 204: Add reference.
Lines 212-239: Add references.
Lines 257-258: You can explain why.
Line 266 (Figure 1): In this figure, the alphabetical order for two different things is distorting. Maybe you can use a, b, c for AMP insertions and you can use 1, 2, 3 or I, II, III for intracellular targets.
Lines 290-291: Where were the structures taken from? Add reference, e.g. PubChem.
Lines 295-314: Add a reference for each drugs
Line 317: How do they differ?
Line 330: “used in combination with nisin” why?
Lines 334-335: Add references for biofilm production and infections and for coronavirus study.
Line 339 (Table 1): fix the parenthesis.
Line 425: add reference (for example doi:10.2174/1381612828666220817163339)
Line 431: “alpha helical” For example, it is not the only essential characteristic.
Line 456: Add reference.
Line 466: “other databases” for example which ones?
Line 491 (Table 3): I would always use the same format for the table.
Lines 525-527: I don't understand. Elaborate on the concept.
Line 573-574: How?
Line 653: Which one? Add the name.
Line 713: Add reference.
Line 723: I would add a sentence about what nanotechnology are, as was done for machine learning
Lines 732-733: How?
Line 822: “in silico” should be in italics
Figure 4: The image is blurred
Figure 6: I would put the figure after line 730 and then begin to talk in detail about the various types of AMP delivery.
Round 2
Reviewer 2 Report
The authors improve the article with the reviewer’s comments